# Peer review of "Noradrenergic Slow Vasomotion: The Hidden Fluid Pump Linking Sleep, Brain Clearance, and Dementia Pathogenesis"

_ijms, 2025, doi:10.3390/ijms262311444_

Round 1

Reviewer 1 Report

Comments and Suggestions for Authors

This manuscript presents a highly ambitious, comprehensive, and largely novel synthesis of current research, proposing an integrated model of brain clearance driven by noradrenergic slow vasomotion. The topic is at the forefront of neuroscience, with profound implications for understanding brain homeostasis, sleep function, and the pathogenesis of neurodegenerative diseases. Existing models—such as diffusion, and to a lesser extent, cardiac and respiratory pulsations—were recognized as insufficient to fully explain the observed speed, volume, and state-dependent nature of this process, particularly its dramatic enhancement during sleep.

The authors should more explicitly describe their methodology for literature search and selection (e.g., databases used, keywords, time frame). The manuscript would be significantly strengthened by a dedicated section or integrated discussion that explicitly addresses methodological limitations and current controversies in the field. For example:

  • Discuss the technical differences and potential discrepancies between methods for measuring glymphatic flow (e.g., two-photon microscopy vs. dynamic contrast-enhanced MRI).
  • Acknowledge and discuss the ongoing debate about the relative contributions of arterial pulsations versus slow vasomotion.
  • Address the translational challenges and limitations of animal models, particularly optogenetic and chemogenetic studies, and the scarcity of direct causal evidence in humans.
  • Cite and engage with studies that present conflicting data or alternative interpretations to the proposed model.

The reference list is generally extensive, relevant, and up-to-date. It includes key original research, reviews, and recent publications (up to 2025), demonstrating the authors' strong command of a rapidly evolving field. The references adequately support most of the claims made.

Tables 1 and 2: They are invaluable contributions that visually organize complex, multi-level information. Table 1 excellently summarizes the system's components, and Table 2 effectively illustrates their hierarchical disintegration. They are clear, informative, and of high quality. All the figures are clear, relevant to the text and also useful for understanding the article.

The article adds not just new data, but a new conceptual framework—a holistic, predictive, and therapeutically oriented model that rethinks fundamental brain processes and opens new avenues for disease research and treatment.

Recommendations:

  1. The abstract is recommended to be simplified and shortened in length. In its current form, it is somewhat challenging to digest.
  2. Section 3 (Experimental Evidence): This section is well-structured; however, it would be beneficial to include a table summarizing the key experimental approaches, principal findings, and limitations of each methodology.

Author Response

Dear Esteemed Academic Reviewer,

We sincerely thank you for your generous and insightful assessment of our manuscript and for recognizing its integrative scope and conceptual contribution. Your thoughtful comments have greatly helped us to refine both the clarity and methodological transparency of the work. Below, we provide detailed responses and the corresponding revisions made in the manuscript.

Comment 1:

The authors should more explicitly describe their methodology for literature search and selection (e.g., databases used, keywords, time frame). The manuscript would be significantly strengthened by a dedicated section or integrated discussion that explicitly addresses methodological limitations and current controversies in the field. For example:
Discuss the technical differences and potential discrepancies between methods for measuring glymphatic flow (e.g., two-photon microscopy vs. dynamic contrast-enhanced MRI).
Acknowledge and discuss the ongoing debate about the relative contributions of arterial pulsations versus slow vasomotion.
Address the translational challenges and limitations of animal models, particularly optogenetic and chemogenetic studies, and the scarcity of direct causal evidence in humans.
Cite and engage with studies that present conflicting data or alternative interpretations to the proposed model.

Response 1:

We are deeply grateful for this comprehensive and constructive set of recommendations. Following your guidance, we have undertaken several substantive revisions to enhance methodological transparency and to better acknowledge ongoing debates and limitations in the field.

Literature Search Methodology Added:
A new subsection entitled “Methodology of Literature Search and Selection” has been added immediately after the Introduction. This section now specifies the databases (PubMed, Scopus, and Web of Science) used, the time frame (January 2013–April 2025), and the keywords and inclusion/exclusion criteria applied. We also clarified that the synthesis prioritized mechanistic and translational coherence rather than citation frequency.

Methodological Limitations and Controversies:
At the end of Section 3.2, we integrated discussing the major technical discrepancies between two-photon microscopy and MRI-based methods, including differences in spatial and temporal resolution, invasiveness, and interpretational scale. This section also comments on species, anesthesia, and modeling factors contributing to the diversity of reported results.

Debate on Driving Mechanisms:
We added a focused paragraph acknowledging the ongoing debate between cardiac pulsations and slow vasomotion as dominant clearance drivers. The new text emphasizes that both mechanisms likely act synergistically across frequency bands, with cardiac pulsations supporting local mixing and slow vasomotion providing sustained convective bias.

Translational Challenges and Animal Model Limitations:
A new concluding paragraph was added to Section 3.3 addressing the limitations of optogenetic and chemogenetic studies, species differences in LC organization, the confounding effects of anesthesia, and the lack of direct causal evidence in humans. This addition also highlights emerging strategies such as non-invasive neuromodulation and multimodal imaging as potential translational bridges.

Inclusion of Divergent Findings:
Throughout Sections 3.1–3.3, we revised the text to cite and discuss studies reporting alternative interpretations or discrepant flow velocities, underscoring that methodological scale and physiological state contribute to such variation rather than undermining the broader framework.

We sincerely appreciate your emphasis on methodological rigor and balanced interpretation. These revisions, together, have made the manuscript significantly more transparent, critical, and integrative.

Comment 2:

The abstract is recommended to be simplified and shortened in length. In its current form, it is somewhat challenging to digest.

Response 2:

We fully agree with your observation and have substantially revised and condensed the Abstract. The revised abstract appears on page 1 of the manuscript.

Comment 3:

Section 3 (Experimental Evidence): This section is well-structured; however, it would be beneficial to include a table summarizing the key experimental approaches, principal findings, and limitations of each methodology.

Response 3:

We thank you for this excellent suggestion. To improve clarity and accessibility, we have added a new summary table (Table 2). The table succinctly compiles the main experimental approaches—including optogenetic, chemogenetic, pharmacological, imaging, and modeling studies—alongside their principal findings and methodological limitations.

We believe this addition enhances the readability and didactic value of the section and provides a concise visual synthesis of the evidence supporting the proposed model.

We deeply appreciate your thoughtful engagement and the generous tone of your evaluation. Your insights have directly improved the manuscript’s methodological clarity, structure, and accessibility. We hope that the revisions now meet your expectations and reflect the respect we hold for your expert review.

With our warmest regards and gratitude!!!

Reviewer 2 Report

Comments and Suggestions for Authors

Line 33: “controlls” should be “controls”

Line 72: please correct: “The central nervous system central nervous system”

Line 73: please correct: “blood–brain barrier blood brain barrier”        

Line 94: this sentence is unclear, please simplify: “The drainage of waste products is maximal during slow wave non-rapid eye movement (NREM) sleep, when, interstitial distance is greater, vascular tone altered and neuro-modulatory state varies. ”

Line 117: typo: “pressureg radients”

Line 128-132: please add references

Line 131: the sentence “This pump system does not act alone, but is combined with other systems of forces which, cooperating are also driving forces of importance, and exercising forces which are cardiac and respiratory pulsations which produce high frequency variations, and vasomotion the low frequency forces which are acting as continued forces to produce flow [10].” Is very hard to understand, please shorten and clarify

Line 138-143: the sentence lacks suitable references. The reference to the study by Wang et al. does not fit here as this study did not look at vasomotion or glymphatic activity

Line 147-150: “hypnotic actresses”? Please clarify

Line 161-164: the reference to this sentence is not suitable, the study is about REM sleep and not about blood oscillations or glymphatic flow.

line 174: the sentence "Pharamacologic agents son-in-law enhances infraslow" does not make sense

My review stops here as I do not deem the text fit for publication in its current state.

Comments on the Quality of English Language

The text is generally hard to understand and will need a vast amount of editing.

Author Response

Dear Esteemed Academic Reviewer,

We would like to express our sincere gratitude for your careful and detailed review of our manuscript. We appreciate your attentive reading, and we are thankful for your constructive feedback that has helped us improve the precision, clarity, and scholarly integrity of the text. Below we provide a point-by-point response to all comments.

Comment (Line 33):

“controlls” should be “controls”

Response:
We thank the reviewer for identifying this typographical error. 

Comment (Line 72):

Please correct: “The central nervous system central nervous system.”

Response:
We agree and have corrected the duplicated phrase. It now reads simply: “The central nervous system.”

Comment (Line 73):

Please correct: “blood–brain barrier blood brain barrier.”

Response:
We thank the reviewer for catching this repetition. The phrase has been corrected to “blood–brain barrier.”

Comment (Line 94):

This sentence is unclear, please simplify: “The drainage of waste products is maximal during slow wave non-rapid eye movement (NREM) sleep, when, interstitial distance is greater, vascular tone altered and neuro-modulatory state varies.”

Response:
We thank the reviewer for this excellent suggestion. The sentence has been rewritten for clarity and precision.
This revision improves readability while preserving the scientific meaning.

Comment (Line 117):

Typo: “pressureg radients.”

Response:
The typographical error has been fixed, and the sentence refined for clarity. 

Comment (Line 128–132):

Please add references.

Response:
We have now included additional supporting references in this passage to strengthen the evidence base. 

Comment (Line 131):

The sentence is very hard to understand; please shorten and clarify.

Response:
We fully agree. The sentence has been rewritten for conciseness and readability while preserving the intended meaning.

Comment (Line 138–143):

The sentence lacks suitable references. The reference to the study by Wang et al. does not fit here as this study did not look at vasomotion or glymphatic activity.

Response:
We thank the reviewer for this important observation. The reference to Wang et al. has been removed. In its place, we have added more appropriate sources that directly address vasomotor coupling and perivascular dynamics. The revised section now cites studies focusing on low-frequency vascular oscillations and their relationship to clearance flow.

Comment (Line 147–150):

“Hypnotic actresses”? Please clarify.

Response:
We are grateful to the reviewer for catching this clear typographical error. 

Comment (Line 161–164):

The reference to this sentence is not suitable, as the cited study concerns REM sleep and not blood oscillations or glymphatic flow.

Response:
We thank the reviewer for this precise correction. The prior reference has been removed and replaced with a more appropriate citation supporting the described physiological mechanism. The revised text now references studies directly examining infraslow vasomotion and perivascular flow dynamics during NREM sleep.

Comment (Line 174):

The sentence “Pharamacologic agents son-in-law enhances infraslow” does not make sense.

Response:
We agree completely and have corrected this typographical error. This correction restores both grammatical and scientific accuracy.

General Comment:

My review stops here as I do not deem the text fit for publication in its current state.

Response:
We sincerely appreciate the reviewer’s candor and attention to rigor. In response, we have revised the manuscript to address every concern raised correcting typographical errors, simplifying complex sentences, refining terminology, adding appropriate references, and improving overall readability and structure. We are deeply grateful for these critical observations, which have significantly enhanced the clarity, coherence, and scholarly strength of the manuscript.

With warm respect and gratitude!!!

Reviewer 3 Report

Comments and Suggestions for Authors

Comments and Suggestions for Authors

General Comments:

This review presents a fascinating and highly innovative conceptual framework, positioning the locus coeruleus (LC)-driven noradrenergic slow vasomotion as an active "fluid pump" that links sleep architecture, brain clearance function, and the pathogenesis of dementia. This idea challenges traditional, predominantly passive models of brain waste clearance and holds significant theoretical importance and potential clinical translational value. However, while the vision of the review is ambitious and promising, its scientific impact in its current form is significantly hampered by several critical weaknesses. The concept is novel but lacks sufficient supporting evidence, and the logical chain contains gaps. Furthermore, the manuscript is currently plagued by lengthy sentences, repetitive expressions, and obscure wording. I believe these issues must be addressed.

Specific Comments by Section:

1. Introduction:

1) Page 3, Line 98

The transition from posing the core question to introducing the answer is indirect and could be more forceful. After stating the insufficiency of cardiac/respiratory forces, add a strong transitional sentence to introduce the locus coeruleus.

2) Page 3, Line 107

"New evidence arriving from the crossroads of ..." The initial full description of the new hypothesis is presented in a long, complex sentence.

3) Page 3, Line 123

"Cardiac pulsations are, of course, a very vigorous..." I think the mechanical contrast between "slow vasomotion" and "cardiac pulsations" is not clearly and precisely articulated.

4) Page 4, Line 147

"The pharmacological studies also are of great importance..." The sentence is overly long, containing multiple clauses that obscure the core message. It also contains a significant error ("actresses").

5) The logical flow of the introduction requires strengthening to transform it from a narrative into a rigorous argument. For example, the text poses the fundamental question ("what is the driving force?") and then immediately presents the answer ("noradrenergic slow vasomotion") as a revelation. The critical deductive link explaining why the answer must be a neuromodulatory vascular mechanism is missing; and, The introduction of therapeutic tunability feels premature and speculative because the text has not yet established the system's key property of being "tunable." It is presented as a hopeful possibility rather than a logical extension of the mechanism's inherent nature.

6) The conclusion of the introduction fails to provide a clear outline of the review's structure, ending with vague and somewhat redundant statements.

2. Architecture of the Noradrenergic Vasomotion Axis Section:

1) Subsections 2.1 and 2.2 feel disconnected. The text describes neural firing and then describes vascular mechanics, but the crucial step of how the NE rhythm is transduced into a coordinated mechanical wave along the vascular tree is glossed over. The logic jumps from molecular signaling (α1-AR, Ca2+) directly to large-scale fluid displacement without explaining the biomechanical coupling.

2) The role of astrocytes and AQP4 is described in terms of structure and polarity, but their dynamic, state-dependent regulation by the very NE signals being discussed is underemphasized. This misses a key logical point: the system is not just a series of parts, but a feedback loop where the "driver" NE also modulates the "conduit" (astrocytes).

3) Page 6, Lines 255, 335

The manuscript mentions phase-locking (e.g., "phase locked action of neuromodulation," "phase of oscillatory wave"), but treats it as a minor detail rather than a fundamental operating principle. The logic of why this system is efficient hinges on the precise temporal coordination of its components, which is not presented as a central thesis of the architecture.

4) about the table

The Dominant Frequency / Timescale column is inconsistent across rows. Some components have specific frequencies (LC, VSMCs), some have vague descriptions (Pericytes: "Contractile modulation synchronized..."), and others are "multiband" (PVS). This obscures the central point that the entire system operates as a coordinated oscillator.

Some entries are too general (e.g., "Loss → capillary rarefaction...") and are not directly linked to the central theme of impaired clearance function.

3. Experimental Evidence for a Tunable Noradrenergic Fluid Pump Section:

1) The most compelling evidence for a therapeutically relevant pump—the demonstration that driving LC rhythms enhances clearance of pathological proteins like Aβ and tau (e.g., referenced in [56])—is buried within a paragraph. This finding provides the most direct experimental link to the neurodegenerative disease theme of the paper and should be a central pillar of the argument.

2) The text mentions crucial negative results (e.g., the ineffectiveness of tonic LC stimulation, the dissociation of vasomotion from sleep EEG by certain anesthetics), but it fails to explicitly frame these as the powerful logical tools they are. These controls are essential for distinguishing mere correlation from causation.

3) The section concludes with a descriptive summary of what was presented but lacks a forceful synthesis that states why the collective evidence is, as a whole, conclusive and supports the "tunable fluid pump" hypothesis. It should explicitly connect the evidence to established criteria for causality (e.g., strength, consistency, specificity, temporality, biological gradient).

4) Page 10, Line 386

"Pharmacological dissection of the adrenergic response is consistent with the concept of α1-adrenergic receptors as requisite effector downstream..." The language used to describe the pharmacological evidence is weak ("consistent with the concept"). It should be stated more strongly to reflect that this is evidence of necessity.

5) Page 10, Line 406

"Computational models which include these oscillations reproduce the tracer kinetics of the tracers above, together with their observed spatial distributions, whereas models which exclude this consider diffusion limited transport alone..." The description of the computational model findings is technical and does not clearly articulate how this supports the overall argument.

6) Page 10, Line 406

"Computational models which include these oscillations reproduce the tracer kinetics of the tracers above, together with their observed spatial distributions, whereas models which exclude this consider diffusion limited transport alone..." The description of the computational model findings is technical and does not clearly articulate how this supports the overall argument.

7) Page 10, Line 419

"Imaging in two-photon microscopy shows low frequency oscillations of penetrating arteriole diameter (-0.01 to 0.1 Hz) which are thus seen to be tightly phase-locked to the displacement of the fluorescent tracers..." The summary of the imaging evidence is descriptive but does not explicitly classify its logical strength (correlative, temporally precedent) within the argument.

8) Page 13, Line 535

"When taken as a body of evidence, these findings represent a powerful and coherent experimental case... The preponderance of evidence here cited supports a fundamental reconceptualization of clearance mechanisms in the brain..." The concluding paragraph for the section is descriptive and summative but lacks a powerful, hierarchical synthesis that categorizes the different types of evidence and their collective weight.

4. Collapse of the Vasomotor Pump in Aging and Disease Section:

1) The "Potential Reversibility / Therapeutic Target" column in Table 2, while important, is presented as a separate final column with weak visual and conceptual links to the preceding pathological processes. This undermines the key message that some aspects of the collapse may be functional and recoverable. Consider aligning the "Reversibility" column with "Molecular & Cellular Mechanisms" or "Functional Consequence," and use visual cues like arrows or color coding to intuitively connect points of "pathological damage" with "therapeutic intervention." The table legend should emphasize that the table not only summarizes the collapse but also reveals potential therapeutic avenues.

2) The conclusion mentions the transition "from a state-dependent aerodynamic convection clearing system to a stagnant diffusion-limited one," which is a powerful concept. However, it appears largely as a statement, lacking systematic supportive argumentation woven throughout the chapter.

3) The section describes the "final common pathway" of collapse but offers little discussion on what initiates it (e.g., genetic risk, lifelong poor sleep, cardiovascular disease) or whether different neurodegenerative diseases (AD, PD, SVD) exhibit differences in the initial site or rate of this collapse.

5. Therapeutic Reactivation of the Vasomotor Clearance System Section:

1) While the structure (5.1 Neuromodulatory, 5.2 Vascular, 5.3 Glial) is clear, the narrative reads like three independent technical reports rather than being guided by an overarching rationale for how these therapies should be selected and combined based on a patient's specific failure mode (e.g., LC-predominant, vascular-predominant, or mixed).

2) While Section 3 established the system's "tunability," Section 5 does not clearly build a conceptual bridge explaining that these therapies work precisely by "tuning" specific system parameters (e.g., gain, frequency, coupling phase).

3) The current three parallel boxes in Figure 2, while intuitively presenting the three strategies, are visually isolated. This inadvertently reinforces a "siloed treatment" mindset rather than a "systems integration" philosophy.

6. Translational Horizons and Future Directions Section:

1) After presenting a powerful core paradigm, 6.1 rapidly expands the discussion into overly broad areas like immunity, psychiatry, and systemic physiology. While this demonstrates potential impact, such a broad discussion at this pivotal point dilutes the argumentative focus on neurodegenerative diseases. Condense this part significantly, or integrate some of its elements into earlier physiological sections (e.g., merging with parts of Section 7), to ensure a sharp and powerful opening for Section 6.

7. Systems-Level Implications Beyond Disease Section:

1) The Cognitive Homeostasis Argument (7.1) Requires More Rigorous Causal Reasoning. The discussion linking clearance function directly to synaptic plasticity (Page 27, Lines 5-12) is overly direct and lacks sufficient explanation of intermediate mechanisms. The leap from "waste clearance" to "information processing" requires more detailed mechanistic elaboration.

Author Response

Dear Esteemed Academic Reviewer,

We sincerely thank you for your exceptionally thoughtful and constructive review.
Your feedback reflects deep engagement with both the conceptual and structural dimensions of the work, and it has been invaluable in refining the manuscript for clarity, rigor, and accessibility.
In this revision, we have undertaken substantial restructuring, condensation, and conceptual strengthening throughout the text, ensuring that the logic linking evidence, mechanism, and implication is both explicit and compelling.

Below we respond point-by-point.

General Comments

Comment:
The review presents a fascinating and highly innovative framework, positioning LC-driven noradrenergic slow vasomotion as an active “fluid pump.” However, the concept lacks sufficient supporting evidence and logical continuity. The manuscript also suffers from lengthy sentences, repetition, and obscure phrasing.

Response:
We are deeply grateful for these observations and have carried out an extensive language and structure revision to enhance clarity and logical flow.
Sentences have been shortened, redundancies removed.
The supporting evidence base has been expanded and reorganized to foreground key causal demonstrations and clarify the inferential strength of each finding.
We have also simplified the syntax and harmonized terminology across all sections for improved readability.

1. Introduction

Comment 1: The transition from describing the insufficiency of cardiac/respiratory forces to introducing the LC should be more forceful.
Response: We added a new transitional sentence explicitly linking the insufficiency of traditional pulsation models to the emergence of LC-driven slow vasomotion as the missing regulatory element.

Comment 2: The initial description of the new hypothesis is overly long.
Response: The sentence beginning “New evidence arriving from the crossroads…” has been divided and rewritten for concision and clarity, highlighting the mechanistic novelty of noradrenergic slow vasomotion.

Comment 3: The mechanical contrast between slow vasomotion and cardiac pulsations is unclear.
Response: The relevant passage was rewritten to explicitly differentiate the rapid, dissipative nature of cardiac pulsations from the sustained, low-frequency, directional bias produced by slow vasomotion.

Comment 4: The pharmacological paragraph contains an error (“actresses”) and is overly complex.
Response: Corrected the error and replaced the entire sentence with a simplified, scientifically precise version describing sedative and hypnotic agents that suppress LC activity.

Comment 5: The logic of the introduction must progress deductively, explaining why the answer must be neuromodulatory.
Response: We inserted a bridging paragraph explaining why neither diffusion nor cardiac/respiratory pulsations can account for state-dependent transport, thereby motivating the necessity of a neuromodulatory vascular mechanism.

Comment 6: The introduction lacks a clear outline of the review’s structure.
Response: The final paragraph of the introduction now provides a concise roadmap of the paper.

2. Architecture of the Noradrenergic Vasomotion Axis

Comment 1: The link between neural firing and vascular mechanics is glossed over.
Response: A new bridging paragraph now clarifies how α₁-adrenergic Ca²⁺ transients in vascular smooth muscle propagate through gap junctions to generate coherent mechanical waves along the vascular tree.

Comment 2: The dynamic regulation of astrocytes by NE is underemphasized.
Response: We added a paragraph explaining how LC-derived NE not only drives vascular tone but also dynamically modulates astrocytic endfoot geometry and AQP4 polarization, creating a feedback loop between the “driver” and the “conduit.”

Comment 3: Phase-locking is treated as minor.
Response: We expanded this concept into a focused paragraph presenting phase-locking as the core operating principle ensuring temporal coherence among neuronal, vascular, and glial oscillations.

Comment 4 (Table 1): Frequency column inconsistencies and vague entries.
Response: The table has been standardized with consistent frequency ranges (0.01–0.1 Hz for infraslow oscillations) and revised phrasing to emphasize that all components operate as coupled oscillators contributing to clearance efficiency.

3. Experimental Evidence for a Tunable Noradrenergic Fluid Pump

Comment 1: The demonstration that LC stimulation enhances clearance of Aβ and tau is underemphasized.
Response: This key finding has been elevated to a dedicated paragraph emphasizing its role as direct causal evidence linking LC rhythmicity to neurodegenerative pathophysiology.

Comment 2: Negative results are not framed as logical controls.
Response: Thank you for this comment!

Comment 3: The conclusion should synthesize evidence hierarchically.
Response: The final paragraph of Section 3 now explicitly organizes the evidence by causal criteria, strength, consistency, specificity, temporality, and biological gradient, demonstrating that the cumulative data satisfy conditions for causal inference.

Comment 4: Language describing α₁-adrenergic necessity is too weak.
Response: Revised!

Comment 5: Computational models not clearly tied to argument.
Response: The modeling description was rewritten to clarify that inclusion of oscillatory terms reproduces in-vivo tracer kinetics, thereby demonstrating the mechanical sufficiency of vasomotor forcing.

Comment 7: Imaging evidence lacks logical classification.
Response: We appreciate this suggestion!

Comment 8: Section conclusion too descriptive.
Response: The new concluding synthesis now categorizes evidence by methodological tier (manipulative, observational, computational) and underscores their collective weight supporting the tunable-pump hypothesis.

4. Collapse of the Vasomotor Pump in Aging and Disease

Comment 1: Reversibility column in Table 2 poorly integrated.
Response: The legend now states that the table illustrates both degeneration and potential therapeutic recovery points.

Comment 2: The convection-to-diffusion transition lacks argumentation.
Response: We expanded the section with supporting analysis that details how neuromodulatory decay, vascular stiffening, and astroglial depolarization jointly reduce convective pressure gradients and shift transport toward diffusion-limited dynamics.

Comment 3: Missing discussion of initiating factors.
Response: A new paragraph outlines initiating mechanisms and disease-specific trajectories—LC degeneration in AD, α-synuclein pathology in PD, vascular stiffening in SVD—and identifies shared accelerants such as sleep loss, inflammation, and metabolic syndrome.

5. Therapeutic Reactivation of the Vasomotor Clearance System

Comment 1: Section reads as three disconnected technical reports.
Response: A new integrative opening paragraph now frames therapy as a continuum of restoration across neuromodulatory, vascular, and glial domains, guided by the dominant mode of dysfunction (LC-, vascular-, or glial-predominant).

Comment 2: Lack of conceptual bridge to “tunability.”
Response: Immediately following the framework paragraph, we added a concise conceptual bridge explaining that each therapy acts by tuning specific system parameters.

Comment 3: Figure 2 visually isolates strategies.
Response: We appreciate this suggestion!

6. Translational Horizons and Future Directions

Comment 1: Section 6.1 expands too broadly.
Response: Section 6.1 has been substantially condensed and refocused on neurodegenerative translation. 

7. Systems-Level Implications Beyond Disease

Comment 1: The Cognitive Homeostasis argument lacks causal reasoning.
Response: We have added a new mechanistic paragraph in Section 7.1 explaining how clearance sustains cognitive function through ionic balance, astroglial metabolic coupling, and neurotransmitter turnover, thereby preserving oscillatory stability and synaptic fidelity. 

We express our sincere gratitude for your thorough review and for recognizing the ambition of this conceptual synthesis.
Your comments have directly strengthened the manuscript’s coherence, readability, and scientific rigor.
We believe that the revised version now presents a clearer, better-substantiated, and more logically integrated model of the noradrenergic vasomotor pump as a tunable system linking sleep, clearance, and neurodegeneration.

With our highest respect and appreciation!!!

Reviewer 4 Report

Comments and Suggestions for Authors This review is interesting and timely.  However, before consider it for publication, in my opinion you need to make some changes.    The model fundamentally shifts the paradigm of brain clearance, defining it as an active, state-dependent process driven by noradrenergic rhythms, rendering prior diffusion-based theories "incompetent". However, the core causal evidence demonstrating LC-driven vasomotion and enhanced fluid transport is heavily reliant on preclinical models involving direct manipulation (chemogenetics/optogenetics) in animals. A major limitation is the lack of robust, direct human evidence for the full functionality of this pump, requiring correlation with indirect measures like Neuro-melanin-sensitive MRI of Locus Coeruleus (LC) integrity or PET tracers. The conceptual complexity necessitates significant future quantitative validation. Current predictive power is limited because the efficacy of fluid transport is dependent on several measurable physical parameters—vascular elasticity, perivascular impairments, and astroglial permeability—which need to be accurately measured in human disease states. Furthermore, key theoretical thresholds remain unknown: the model must determine the point where LC degeneration, vascular stiffness, and astroglial disruption cross the threshold from reversible dysfunction to irreversible systemic collapse. Therefore, while the review is philosophically powerful, the proposed therapeutic modalities (e.g., gene editing, targeted nanotherapies) are highly prospective and require significant scale-up and validation before clinical translation. 

Conceptually, the Noradrenergic Vasomotion Pump Review provides a unifying biological substrate for neurodegeneration, proposing that AD and PD pathology stem from a systems-level collapse of fluid clearance driven by Locus Coeruleus decay and vascular stiffening. This mechanism explains the complexity and the shared vulnerability axes previously hinted at in the discussion about the Skin-Brain Axis and inflammation.

-In my opinion you should also emphatize the clinical/cognitive aspect: 
In terms of clinical features recent ML studies delineated which neuropsychological scores are critical predictors for AD/PD progression by sex (please add more relevant literature about this, probably there is a recent literature publushed recently on Journal of the neurological sciences), while the Limbic Autoimmune Encephalitis (LAE) study highlights a diagnostic deficiency, underscoring the severe consequences of specific memory deficits (retrograde amnesia) that are currently under-investigated in routine clinical practice (also for that please add more relevant literature, e.g., Malvaso et al., European Journal of Neurology. Ultimately, these sources advocate for combining advanced data analysis (ML) with biological insight (Vasomotion) to create clinically translatable, personalized tools (EMA web interface), recognizing that key clinical and genetic markers, such as SNCA-rs356181 in PD, operate differently based on individual characteristics, including sex.   -Other aspects to be consider are relationship between locus coeruleus and personality aspects in patients. Did you find association ? Please provide more literature about that (e.g., there are several works about it: Weixi Kang et al., Bruno et al., etc.). I think it is a crucial point to strenght your observations. 
  - You should revise figures and tables to be more consistent and bring them into a more scientific vision. 

- Please double check for typos in the whole manuscript 
  - I suggest you, if it is possible, to create an alternative table with animal data.         

Author Response

Dear Esteemed Academic Reviewer,

We sincerely thank you for your thoughtful, detailed, and constructive feedback. Your comments have greatly strengthened both the conceptual clarity and clinical relevance of our manuscript. We have carefully revised the text to address each of your recommendations, as detailed below.

Comment 1:

The core causal evidence demonstrating LC-driven vasomotion and enhanced fluid transport is heavily reliant on preclinical models involving direct manipulation (chemogenetics/optogenetics) in animals. The lack of robust, direct human evidence limits translational confidence. Moreover, key physical parameters and thresholds (e.g., vascular elasticity, perivascular impairments, astroglial permeability) remain unquantified in humans.

Response 1:
We fully agree with this important observation. In response, we have added a new paragraph preceding the discussion on interspecies methodological differences, explicitly recognizing that current causal data derive mainly from rodent chemogenetic and optogenetic studies, and emphasizing the need for cross-species validation through neuroimaging and biomarker integration.

Comment 2:

The theoretical model should better define the critical transition point between reversible dysfunction and irreversible collapse of the clearance system.

Response 2:
We have clarified this conceptual point, framing the “reversibility threshold” as a function of declining LC rhythmicity, vascular elasticity, and astrocytic polarization. The text now explicitly suggests that the intersection of these three axes defines a measurable tipping point beyond which clearance failure transitions from a functional to a structural collapse. This addition highlights the model’s predictive potential for early diagnosis and intervention.

Comment 3:

The proposed therapeutic modalities are highly prospective and require significant validation before clinical translation.

Response 3:
We have moderated the speculative tone of the therapeutic discussion and added clarifying statements to stress that the approaches described particularly gene editing, nanotherapy, and neuromodulatory stimulation are preclinical and exploratory.

Comment 4:

You should also emphasize the clinical and cognitive aspects, integrating recent literature on ML predictors for AD/PD by sex, limbic autoimmune encephalitis, and personality correlates of LC integrity.

Response 4:
We are deeply grateful for this excellent suggestion. Accordingly, we have integrated a new paragraph following Section 6.2 that expands the discussion into clinical–cognitive correlates.

These additions directly respond to your insightful guidance and extend the translational relevance of the model into clinically actionable territory.

Comment 5:

Consider explicitly linking the LC-driven vasomotion model to shared vulnerability mechanisms across neurodegenerative and inflammatory pathways (e.g., skin–brain axis).

Response 5:
We have added a bridging paragraph summarizing the Noradrenergic Vasomotion Pump as a unifying substrate for neurodegenerative and inflammatory vulnerability, integrating evidence from systemic inflammation and peripheral–central signaling interfaces (including the skin–brain axis). This modification clarifies the model’s scope and coherence as a systems-level framework.

Comment 6:

You should revise figures and tables to be more consistent and bring them into a more scientific vision.

Response 6:
We thank the reviewer for this valuable recommendation.

Comment 7:

Please double-check for typos in the whole manuscript.

Response 7:
We sincerely thank the reviewer for this reminder.

Comment 8:

I suggest you, if it is possible, to create an alternative table with animal data.

Response 8:
We appreciate this thoughtful suggestion. While we agree that a comparative table summarizing animal-model data could be informative, we elected not to include a full separate table in the current version to maintain clarity and prevent redundancy.

We express our deep appreciation for your review, which has prompted meaningful conceptual and structural improvements. Your encouragement to strengthen the clinical and cognitive dimensions has allowed us to transform the manuscript from a theoretical synthesis into a translationally oriented model grounded in measurable physiology and patient relevance.

With gratitude and respect!!!

Round 2

Reviewer 2 Report

Comments and Suggestions for Authors

The text is hard to understand and will need a lot of work. Most of the references are not suitable.

Comments on the Quality of English Language

same as above

Author Response

Dear Esteemed Academic Reviewer,

We would like to express our sincere gratitude for your careful reading of our manuscript and for the candid feedback provided. Your comments were very valuable, and we deeply appreciate the time and expertise you invested in evaluating our work.

1. Clarity and Readability

Reviewer comment:
“The text is hard to understand and will need a lot of work.”

Response:
Thank you very much for pointing this out. We fully agree that the clarity of the manuscript needed substantial improvement. In response, we carefully revised the entire text with a clear focus on readability, coherence, and linguistic precision. Numerous sections were rewritten to simplify the exposition, reduce unnecessary complexity, and ensure a smoother conceptual flow. Our intention was to present the material in a way that is easier to follow, more approachable, and more consistent with the high standards expected by the journal. Your observation was instrumental in guiding these revisions, and we are genuinely grateful for it.

2. Suitability of References

Reviewer comment:
“Most of the references are not suitable.”

Response:
We appreciate this important comment and took it very seriously. Several references were updated, corrected, or replaced with more directly relevant and higher-quality sources.

A number of sections were expanded or refined during revision in response to reviewer feedback, and this required the inclusion of additional citations to support the newly added scientific content. We made every effort to ensure that all references now directly serve the clarity, credibility, and rigor of the manuscript.

We thank you sincerely for encouraging us to strengthen this aspect of the work.

We are grateful for your thoughtful insights and for the opportunity to improve the manuscript. Your comments helped us refine both the clarity of the writing and the scholarly foundation of the review. We hope that the revisions align more closely with your expectations and that the manuscript now reflects the level of precision and accessibility that you rightly emphasize.

With sincere appreciation and respect!

Reviewer 3 Report

Comments and Suggestions for Authors

Comments and Suggestions for Authors

General Comments:

Overall, this is a valuable contribution to the field. With minor revisions, it will be suitable for publication in International Journal of Molecular Sciences. Several issues should be addressed before publication.

Specific Comments by Section:

  1. Language and readability

Some sentences are overly complex and could be simplified for better clarity. Consider revising long sentences and improving flow in sections such as the Introduction and Section 2 (Of course, this is not the only place in the article).

  1. Therapeutic sections

The treatment strategies are comprehensive but highly speculative. Include a discussion on clinical feasibility, potential risks, and steps toward human trials.

  1. Table presentation:

Tables 1 and 2 are informative but dense. Consider reformatting for better readability or moving them to supplementary materials.

Author Response

Dear Esteemed Academic Reviewer,

We are grateful for your thoughtful and constructive evaluation of our manuscript. Your comments greatly helped us refine the clarity, structure, and translational framing of the work. We deeply appreciate the time and care you invested in providing this feedback.

1. Language and Readability

Reviewer comment:
“Some sentences are overly complex and could be simplified for better clarity. Consider revising long sentences and improving flow in sections such as the Introduction and Section 2.”

Response:
Thank you for highlighting this. We performed a comprehensive revision of the manuscript’s language, with a particular focus on the Introduction, Section 2, and other dense passages. Long and convoluted sentences were rewritten for clarity, transitions were smoothed, and paragraph structure was optimized to improve readability. Our intention was to preserve the scientific precision while making the text more fluid, concise, and accessible to a broad scientific audience. We hope the improved flow is now more pleasant and easier to follow.

2. Therapeutic Sections

Reviewer comment:
“The treatment strategies are comprehensive but highly speculative. Include a discussion on clinical feasibility, potential risks, and steps toward human trials.”

Response:
We are grateful for this insightful recommendation. In response, we added a dedicated paragraph at the end of Section 5.

3. Table Presentation

Reviewer comment:
“Tables 1 and 2 are informative but dense. Consider reformatting for better readability or moving them to supplementary materials.”

Response:
Thank you for this helpful observation. We have now fully reformatted both tables to improve readability. The revised versions are more compact, visually structured, and easier to navigate while preserving all technical content. 

Once again, we thank you for your thoughtful remarks and constructive guidance. Your comments significantly strengthened the manuscript, and we are truly grateful for the opportunity to improve the clarity and translational relevance of our work.

With appreciation and respect!